# Potential Roles of Soil Microorganisms in Regulating the Effect of Soil Nutrient Heterogeneity on Plant Performance

**DOI:** 10.3390/microorganisms10122399

**Published:** 2022-12-03

**Authors:** Michael Opoku Adomako, Sergio Roiloa, Fei-Hai Yu

**Affiliations:** 1Institute of Wetland Ecology & Clone Ecology, Taizhou University, Taizhou 318000, China; 2BioCost Group, Department of Biology, Faculty of Science, Universidade da Coruña, 15071 A Coruña, Spain

**Keywords:** arbuscular mycorrhizal fungi (AMF), clonal plants, foraging mechanism, nutrient acquisition strategy, plant–soil microbe interactions, selective placement

## Abstract

The spatially heterogeneous distribution of soil nutrients is ubiquitous in terrestrial ecosystems and has been shown to promote the performance of plant communities, influence species coexistence, and alter ecosystem nutrient dynamics. Plants interact with diverse soil microbial communities that lead to an interdependent relationship (e.g., symbioses), driving plant community productivity, belowground biodiversity, and soil functioning. However, the potential role of the soil microbial communities in regulating the effect of soil nutrient heterogeneity on plant growth has been little studied. Here, we highlight the ecological importance of soil nutrient heterogeneity and microorganisms and discuss plant nutrient acquisition mechanisms in heterogeneous soil. We also examine the evolutionary advantages of nutrient acquisition via the soil microorganisms in a heterogeneous environment. Lastly, we highlight a three-way interaction among the plants, soil nutrient heterogeneity, and soil microorganisms and propose areas for future research priorities. By clarifying the role of soil microorganisms in shaping the effect of soil nutrient heterogeneity on plant performance, the present study enhances the current understanding of ecosystem nutrient dynamics in the context of patchily distributed soil nutrients.

## 1. Introduction

In terrestrial ecosystems, plants interact with a myriad of soil microbial communities that lead to the establishment of interdependent relationships [1], which drive plant community productivity [2], belowground biodiversity, and ecosystem multifunctionality [3,4,5]. These interactions are crucial for many aspects [6,7], including the nutrient acquisition of plants from heterogeneously distributed microsites. The responses of plants to spatially, heterogeneously distribute soil nutrients require a specialized physiological strategy commonly referred to as the root foraging mechanism, i.e., the proliferation of roots in nutrient-rich microhabitats [8,9] and microbially mediated mechanisms via a plant–microbe symbiotic relationship to ensure the effective acquisition of soil nutrients [10]. For example, arbuscular mycorrhizal fungi (AMF) are essential to these plant–soil microbe interactions [11]. Whilst both soil microorganisms and soil nutrient heterogeneity are known to influence plant performance, community productivity, and competitive interactions [2], little is known about the potential impacts of soil microorganisms on the effect of soil nutrient heterogeneity on plant growth (see [12,13]).

Soil nutrient heterogeneity may be defined as the variability in the distribution of the available soil nutrients within the soil matrix in a given microhabitat [14,15]. The unequal distribution of available nutrient resources may originate from various sources, such as uneven distribution and decomposition of litter in the soil [16,17], differences in the parent material during weathering processes, topography, climate, and differences in the availability of microorganisms [14,15]. Spatial heterogeneity of soil nutrients is ubiquitous in the ecosystem and plays a critical role in the growth of the individual plant [13,18,19], population structure [20], community productivity [21,22], intraspecific and interspecific interaction, and species coexistence [19,23,24,25]. One mechanism underlying such an impact in plants is the capacity of the root foraging response by the roots in nutrient-rich microsites [9,26]. However, the size or scale of the heterogeneity (i.e., patch scale) [27] and the differences in nutrient level (i.e., patch contrast) are the two major characteristics of soil nutrient heterogeneity that can influence plant growth [28]. A common reason is that the responses of plants to soil nutrient heterogeneity via the foraging mechanism are highly dependent on both patch scale and patch contrast [29]. Previous studies have shown that a plant can respond to soil nutrient heterogeneity at one scale but may be unresponsive at another [8,30,31]. Therefore, the patch size and contrast of soil nutrients can determine the effect of spatial heterogeneity on plant growth [32,33]. Nevertheless, the activities of soil microorganisms may influence the distribution of soil nutrients irrespective of the patch size or patch contrast [13,34], thereby altering the effect of soil nutrient heterogeneity on plant performance.

The soil microbial communities constitute a diverse group of microorganisms whose activities can positively or negatively impact the growth and productivity of plants [2,6]. In effect, soil microorganisms directly influence plant growth by forming a mutual (symbiotic) or pathogenic relationship with the roots and, through the free-living microorganisms (non-symbiotic) that are indirectly capable of switching the rate of nutrient supply to plants [2]. Common among these root-associated microorganisms are AMF, which have the potential to supply limiting nutrients such as soil phosphorus (P) to its host in return for carbon [35]. This AM fungal-plant association is the most ancient and abundant relationship in the terrestrial ecosystem [36,37]. Moreover, the N-fixing bacteria provide the highest quantity of soil N for plant community productivity in most ecosystems, especially in plant communities dominated by legumes [2,38]. This N represents about 20% of the total N needed by plants annually [39,40]. Indirectly, there is a considerable number of microorganisms that are N-fixers but are not in a direct symbiotic relationship with any vascular plant, e.g., free-living N_2_ fixers, cyanobacteria that fix N_2_ via its symbiotic association with lichens and bryophytes [39]. Additionally, some filamentous actinomycetes, such as *Frankia*, are known to fix N_2_ as free-living or when in a symbiotic relationship with nonluminous vascular plants [39,41]. Despite these pieces of accumulating evidence that soil microorganisms can influence the growth of plants, knowledge of how these belowground communities mediate plant nutrient acquisition in patchily distributed soil nutrients in the terrestrial ecosystem is still limited.

This review highlights the ecological importance of soil nutrient heterogeneity and microorganisms and discusses plant nutrient acquisition mechanisms in heterogeneous soil. We also examine the evolutionary advantages of nutrient acquisition via soil microorganisms in a heterogeneous environment. Additionally, it highlights a three-way interaction between plants, patchy nutrient distribution, and soil microbial communities. Finally, we propose areas for future research priorities.

## 2. Ecological Importance of Soil Nutrient Heterogeneity

It is well known that the pattern of distribution in soils of essential nutrients for plant growth always displays variation at a range of different spatial scales. Over the years, significant ecological investigations on soil nutrient heterogeneity have been carried out on individual plant growth [19,21], population and community productivity [20,42], and ecosystem nutrient dynamics [43]. Across all findings, evidence is accumulating that the most significant influence of biodiversity on the terrestrial ecosystem functioning is dependent on the heterogeneously distributed soil nutrient [43]. The main reasons underlying such species’ response to the spatial distribution of soil nutrients include enhancing competitive, interspecific interactions in plant communities. As soil nutrients are ubiquitous in space and time, competition for available soil nutrients may depend on plant functional group identity and diversity [43,44]. For example, Garcia Palacios et al. found species-specific responses of three functional groups (grasses, legumes and non-legumes forbs and a combination of them) to heterogeneously distributed soil nutrients in terms of plant biomass accumulation. 

Moreover, plants have varying foraging strategies for nutrient uptake in a heterogeneous environment. These strategies include the selective placement of roots into high-nutrient patches, changes in biomass allocation [23,25,45], and the modification of nutrient uptake capacity [46]. Such differences in the belowground nutrient acquisition are likely to induce disparities in growth and biomass accumulation, thus resulting in the exclusion of the slow-growing plants from the ecosystem by the fast-growing plants [23]. In this way, soil nutrient heterogeneity can modulate the composition and structure of plant communities.

Additionally, the patch scale and patch contrast are inherent features of soil nutrient heterogeneity [21,30,47]. Therefore, plant responses to unevenly distributed soil nutrients may rely on species’ sensitivity to a specific patch scale [48]. Thus, another species may be successful in a multi-species environment if one species fails to acquire nutrients from a specific patch scale. It is worth noting that patch contrast (i.e., the level of nutrient availability) may influence nutrient acquisition and plant demand for a particular nutrient type that is likely to limit its growth [49]. Moreover, the level of nutrient availability, in some cases, compels some plant species with little nutrient demand to employ avoidance mechanisms [20] while others aggressively proliferate their roots in this environment [50]. Such complementary responses of individual species to heterogeneously distributed nutrients may promote species coexistence and the efficient use of ecosystem nutrients.

Lastly, soil nutrient heterogeneity modulates ecosystems’ response to global environmental change. The global-change drivers interacting with soil nutrient heterogeneity include elevated nitrogen deposition, altered rainfall patterns, and increased atmospheric CO_2_ [51]. The beneficial effects of soil nutrient heterogeneity on plant performance in species and population studies mainly occur under a high-nutrient availability [26,27,29]. For example, Maestre and Reynold [52] observed a significant increase in the shoot biomass of an experimental grassland community owing to the interaction between soil nutrient heterogeneity and nutrient availability. Likewise, it is reported that soil nutrient heterogeneity regulated the impact of elevated atmospheric CO_2_ on grassland nutrient use efficiency [42]. One underlying mechanism expected to stimulate such interaction within the plant community level is the presence of co-occurring species, which may lead to differences in plant responses to soil nutrient heterogeneity, suggesting that the plant-nutrient uptake capacity in a heterogeneous environment will likely determine soil heterogeneity as a modulator of ecosystem responses to a surge in atmospheric CO_2_. 

## 3. Ecological Impacts of Soil Microorganisms

The activity of soil microbial communities is considered the lifeline of global ecosystem productivity and sustainability [53,54]. However, plant response to the functional diversity of these unseen soil communities is a crucial determinant of the structure and composition of plant communities. The essential roles that are very critical to the functioning of both above- and belowground plant productivity include, but are not limited to, molecular transformation (organic matter transformation and inorganic transformation), nutrient cycling (nitrogen fixation and carbon cycling), and soil transformation (formation and development of soil).

### 3.1. Molecular Transformation (Organic Matter and Inorganic Transformation) 

Plants require inorganic N for growth [55,56,57]; however, a greater proportion (*c*. 95%) of nitrogen (N) in terrestrial ecosystem soil is in an organic form as amines and amides [58]. Therefore, access to this inorganic N requires a complete decomposition and mobilization of these molecules [55]. Plants depend on soil microorganisms for the enzymatic liberation of soil N and make it available for use [59]. The mycorrhizal fungi are an essential group of soil microorganisms that enhances soil N acquisition by plants. AMF and ectomycorrhizal fungi (ECM) are two significant players in plant–mycorrhizal interaction in the ecosystem [60,61]. These microorganisms are involved in the assimilation of amino acids and sugars [58] and the degradation of proteins [7], thereby making the embodied N available to their host. For example, saprophytic fungi and some soil bacteria can equally immobilize soil N from organic materials; however, ECM can use lignocellulolytic enzymes to break down the organic molecule and release the inorganic N to the host [58]. Mycorrhizal fungi, therefore, modify plant inorganic soil N uptake by increasing the absorbing surface area and the volume of soil exploited by the fine root system of plants [58]. Such soil microbial-assisted plant inorganic N acquisition is vital for deciphering the soil biogeochemical cycles and ecosystem sustainability.

### 3.2. Nutrient Cycling (Nitrogen Fixation and Carbon Cycling)

Nutrient availability determines the plant communities’ above- and belowground productivity, ecosystem stability, and nutrient dynamics [54,62,63]. As a result of intermittent, essential nutrient (e.g., N and P) deficiencies and limitations in natural systems [64], plant–microorganism interaction becomes necessary to offset such nutrient challenges [1,65,66]. Thus, soil microorganisms supply their host with the requisite N and P in return for soil carbon [37]. One such plant–soil microbe interaction is the leguminous–rhizobia interaction that replenishes N naturally into the soil (referred to as biological nitrogen fixation) [67]. 

Previous studies have indicated that nitrogen fixation accounts for about 97% of soil N [68,69,70]. Other microbial partners (e.g., Burkholderia and Cupriavidus) can nodulate legumes and fix soil N [71,72]. Besides the well-known legume-rhizobia, other free-living microorganisms nodulate with a diverse group of angiosperms to fix N, also referred to as asymbiotic nitrogen fixation [69]. Recent evidence suggests that asymbiotic nitrogen fixation constitutes a crucial N input to terrestrial ecosystems with limited symbiotic partners [73].

### 3.3. Soil Formation

The formation and fertility of the soil from parent materials involves physical, chemical, and biological processes [74,75]. The properties of the soil formed are influenced by factors like topography, time, climate, and parent material, as well as the plants and soil microorganisms present [75]. However, the nutrients available for plant growth highly depend on the microorganism’s type, functional identity, and microbial biomass. Therefore, the types, functional identity, and microbial biomass play a vital role in nutrient transformation, nutrient storage, and nutrient cycling [76], which can be a reliable indicator of the soil quality, the stability of the belowground food web, and the ecosystem functioning. For example, in an earlier study, Taunton et al. [76] observed that the soil microbial communities were responsible for regulating phosphate and lanthanide distribution during weathering and soil formation.

## 4. Mechanisms of Plant Nutrient Acquisition in Heterogeneous Soil

Generally, all living organisms include among their vital functions the search for and exploitation of resources they have found in what is known as ‘foraging’ behavior. Nutrients in the terrestrial ecosystem soil are ubiquitously distributed, which ultimately explains the varying mechanisms by which plants acquire the needed nutrient resources for growth and reproduction [77,78]. Plants increase the uptake of such nutrients by altering their physiology and morphological structure, as well as employing some members of the soil microbial communities [77]. In this review paper, we will limit our discussion to three nutrient acquisition mechanisms in terrestrial ecosystems. 

### 4.1. Root Foraging Mechanism

The strategic proliferation of roots for a greater uptake of spatially heterogeneous soil nutrients for growth is considered a survival or an adaptive mechanism in plants [79]. Plants with a strong capacity to acquire such patchy nutrients increase growth, productivity, and fitness [26]. In a patchily distributed nutrient environment, plants modify nutrient uptake efficiency by increasing biomass allocation or the length of their fine roots system in nutrient-poor patches [80]. For example, in a recent study, Liang et al. [65] demonstrated that plants significantly increased root biomass and AM colonization in heterogeneous environments, promoting nutrient acquisition in nutrient-poor soil. However, species specificity plays a significant role in the magnitude and quantity of plant foraging performance [45,65,81], suggesting that such plastic responses can be constrained by certain factors. Weiser et al. [45] enumerated two of these possible factors that can hamper plant foraging performance: (1) differences in growth rate that may define the total size of the root system and (2) the differences in the processes that may define the characteristics of a species. 

Indeed, we add that the growth form of a plant could also constitute a factor that may hinder foraging performance. For example, clonal and non-clonal species have been found to differ accordingly in terms of foraging ability and foraging precision (i.e., the proportion of root biomass in nutrient-rich zone relative to nutrient-poor zones) in nutrient-rich hotspots [45,47,82]. Despite these observed adverse effects, the root foraging strategy remains the sole mechanism plants can rely on for effectively exploiting patchily distributed nutrient resources (see Figure 1).

### 4.2. Clonal Foraging Mechanism

The foraging mechanism is a general plant strategy to explore patchy soil nutrients. However, clonal plants, i.e., plants with a unique capacity to produce potentially physiologically independent vegetative individuals, commonly referred to as ramets, can selectively place their ramets into nutrient-rich microsites for greater uptake, survival, and increased fitness. Foraging for resources is very efficient in clonal plant species [83]. Thus, as clonal plants expand, they cope with environmental heterogeneity and show the ability to selectively colonize and exploit resource-rich patches and avoid unfavorable ones [84]. In addition, ramets in a clonal system can remain vegetatively attached to the parent and receive nourishment support via physiological or clonal integration until they are well-developed to take up their nutrients [9,26]. With multiple generations of ramets attached to a single clone, resource limitations within the ambient environment could induce deleterious impacts on the growth and productivity of the clone [85]. Therefore, nutrient foraging in clonal plants is a strong physiological or adaptive response to spatially distributed soil nutrients [8,86]. In effect, foraging in clonal plants is in two levels—at the parent and ramet levels, which surpasses the one-level foraging of non-clonal plants. The clonal foraging mechanism will likely ensure the efficient exploitation of patchily distributed soil nutrients. 

Previous studies have demonstrated, with different clonal species, that ramets located in high-nutrient patches provide support, e.g., exchange resources and signals to increase the growth and survival of other connected ramets found in low-nutrient patches [8,20,21,85,86]. Resource sharing between connected ramets within clones is especially likely to improve performance when the ramets experience contrasting levels of resource availability [87,88]. However, such responses could also be species specific, as some clonal species differ primarily in growth forms, genotypes, and stands formation [23,44]. Moreover, physiologically integrated ramets can respond plastically to the local conditions they experience, and more interestingly, ramets can share resources and information through their connections [83]. This allows the development of what is known as the division of labor in clonal plants, with a ramets specialization for the uptake of more locally abundant resources and the subsequent increase of the overall clone’s performance in habitats where connected ramets experience contrasting availabilities of different resources [89,90]. Thus, each ramet of the clonal system can be considered as a point for the acquisition of soil resources.

Despite the extensive recognition of the importance of foraging in clonal plants, adverse effects or consequences, such as intraspecific competition within or between genotypes [25,91], transfer of diseases and infections [92], and penalties for parents for establishing offspring ramets in harsh environments, such as heavy metal contaminated zones [93], are uncommon.

### 4.3. Soil Microbe-Mediated Mechanisms of Plant Nutrient Acquisition in Heterogeneous Habitat

As sessile organisms, plants respond to varying environmental cues, such as an unequal distribution of soil nutrients, by developing unique mechanisms to explore and influence their growth and fitness, as discussed above. However, the soil microbial communities, unlike plants, have the potential to react to varied environmental challenges by moving beyond their immediate stress. Therefore, soil microorganisms can enhance plant nutrient acquisition in heterogeneous habitats in the following interactive ways.

#### 4.3.1. Exploitation of Patchy Nutrients via AMF Hyphal Network

The plant-mycorrhizal fungi association is one of the most meaningful plant–soil microbe interactions that have received significant consideration in the scientific frontiers. These interactions thrive on mutual grounds as mycorrhizal fungi enhance plant nutrient acquisition, especially soil N and P, in exchange for soil carbon (soil C) [94,95]. More importantly, AM fungi promote plant nutrient acquisition and water uptake to increase biomass accumulation in a heterogeneous environment via their extensive, extra radical hyphal networks [96,97,98]. Some earlier studies reported that the extensive hyphal networks of the mycelial serve as a conduit for the exportation of nutrients from nutrient-rich microsites to increase the growth of ramets occupying nutrient-poor microsites [12,99]. In a more recent study, Jiang et al. [96] found that AM fungi transported phosphate-solubilizing bacteria to organic patches to facilitate the mineralization of organic P. These phosphate-solubilizing bacteria constitute about 40% of bacteria in the soil [100]. Barrett et al. [101] also found that *Glomus hoi* inoculated into *Plantago lanceolata* promoted the acquisition of soil N from organic patches (see also [102,103]). Similarly, Li et al. [98] reported that the AM fungi *Glomus etunicatum* stimulated the root growth and uptake of the N, P, and K of *Biden pilosa* L., significantly increasing the overall performance in karst soil. Nevertheless, the effects of soil microorganisms in patchy nutrient exploitation have varied depending on the identity of the microorganism present in the soil (see Table 1).

Accumulated evidence has proven that plants prefer to acquire nutrients from patchily distributed microhabitats via fungal networks rather than using roots [106,111]. One key reason underlying such AM fungi-mediated nutrient acquisition is that it is ecologically cost-effective and efficient compared to the high metabolic expenditure involved in the root extension needed to transport a similar quantity of soil nutrients [77,111,112]. It is worth noting that plant–soil microbe interactions can reduce the beneficial effects of soil nutrient heterogeneity on plant growth and productivity, as large quantities of soil nutrients is exported from high-nutrient patches to ramets growing in low-nutrient patches. 

#### 4.3.2. Organic Matter Mineralization and Redistribution in the Soil Matrix

Alternatively, the soil microbial communities can modify plant nutrient acquisition in heterogeneous environments by increasing the decomposition and mineralization of the incorporated organic materials in the soil. These mineralized organic substrates are temporarily immobilized into the soil microbial biomass, stabilizing and reducing nutrient losses via leaching [113]. Previous studies have demonstrated that soil microorganisms represent a significant nutrient pool due to their capacity to outcompete plants for more soil nutrients [113,114]. However, plants can only access these immobilized soil nutrients via microbial turnover [113,114]. Therefore, soil microbial communities can enhance the mineralization of organic matter and redistribute the immobilized soil nutrients across patches [96,115]. Nevertheless, this nutrient acquisition mechanism comes with an opportunity cost, as the presence of soil microbial communities can reduce the contrast between nutrient-rich and nutrient-poor zones and weaken the beneficial effect of the spatial heterogeneity of soil nutrients (see Figure 2, Table 2 [13]).

The soil microbial community’s interaction with AM fungi facilitates litter decomposition [116]. The AM fungi are not likely to induce a direct influence on the decomposition processes [56]. It, however, depends on the saprophytic soil microbial communities to break down the complex organic molecules to release inorganic N, which can then be transported into the AM hyphal network [116]. As a confirmation, Jiang et al. [96] recently reported that AM fungi transported phosphate-solubilizing bacteria to organic patches to facilitate the mineralization of soil organic P, suggesting that mycorrhizal fungi serve as a conduit for transporting the agents of decomposition and processed materials to their host.

#### 4.3.3. Homogenization of Patchy Nutrients via Enzymatic Activities

Nutrients in the terrestrial ecosystem occur in different chemical forms and access to these nutrient forms by plants requires special enzymes. The soil microbial community represents a major source of these enzymes [117], which are patchily distributed in organically bound molecules [115,118]. Under such conditions, specific root-associated soil microorganisms must mineralize these organic-bound molecules into a more available form for plant uptake. Transformations of organic substrates, such as N, C, and P, as well as sulfur (S), increase the soil communities’ enzymatic activity [54]. As enzymatic activity increases, these microbial activities will promote the availability and access to the most limiting nutrients to support the metabolic demands of both the soil microbial communities and plants. The mineralized nutrients become more porous and easily transported by the extensive hyphal networks of AM fungi [61], which tend to modify the absorbing surface area by acting as an extension of the plant root system [119]. However, releasing these limiting nutrients into forms that plants can efficiently utilize seems beneficial to general plant growth. Nevertheless, as the patchily distributed soil nutrients become homogenized via enzymatic activity, the positive effects of soil nutrient heterogeneity on plant growth and productivity can be altered. 

Overall, this microbially, mediated nutrient acquisition from heterogeneous environments seems advantageous to ramets located in low-nutrient microsites, as the mechanism of exporting nutrients between the low- and high-nutrient patches usually occur at the expense of the positive effects of soil nutrient heterogeneity on plant growth. So far, very few studies have observed such an impact; therefore, more empirical studies are needed to deepen our understanding of such impact at both individual plant and community levels. Moreover, under the ongoing global environmental change, elevated atmospheric nutrient deposition may cause an imbalance in nutrient availability, such as N and P limitations, which can lead to root deaths and belowground nutrient losses [113]. However, previous studies have indicated that soil microorganisms are strong competitors for nutrients compared to plants [99,113]. Therefore, soil microorganisms reduce the imbalanced nutrient effects on plants by temporarily immobilizing high quantities of nutrients into their biomass, representing a vital nutrient pool for plants via microbial turnover.

## 5. Evolutionary Advantages of Microbially Mediated Nutrient Acquisition in a Heterogeneous Environment

Both plants and the soil microbial community employ many nutrient acquisition strategies to exploit the available nutrients within their environment. Mycorrhizal fungi associate with approximately 80% of all vascular plants in the terrestrial ecosystem [95,120]. These plant–AM fungi interactions evolved long ago due to the abundance of beneficial plant exudates. Gaining access to these exudates then compelled the fungi to penetrate the living tissues of plants for more of these exudates, without causing harm to their host [121]. As plants became a safe habitat providing shelter and energy, the life of fungi became highly dependent on the host plant. The exchange of limited nutrient resources between the early plants and fungi resulted in a mutual relationship because fungi are required to replace the leaked (as exudates) essential mineral nutrients from plants [121]. More importantly, mycorrhizal nutrient acquisition compared to plant roots is (1) economically cost-effective, (2) occupies large volumes and soil spaces relative to roots, and (3) is limitless in nutrient acquisition.

### 5.1. Ecologically Cost-Effective

As explained in the Section 4.3.1 “*exploitation of patchy nutrients* via *AM fungal network,*” previous studies have reported that vascular plants that form a symbiotic relationship with mycorrhizal strongly prefer to acquire soil inorganic or mineral nutrients compared to roots [99,113]. One reason underlying such microbially-mediated nutrient exploitation preferences is due to its ecologically cost-effectiveness [104,120]. Thus, exporting the same quantity of soil nutrients via the extensive hyphal networks is cheaper than the metabolic cost involved in root extension or foraging [77].

### 5.2. The Wider Proliferation of Mycelial Networks

The effectiveness of plant nutrient acquisition decreases as the availability of soil nutrients decreases [122], contrary to mycorrhizal’s increased nutrient uptake in low-nutrient microsites. For example, plants rely more on mycorrhizal nutrient uptake when soil inorganic P is low for these reasons: (1) mycorrhizal fungi can proliferate their mycelial networks widely beyond the depleted root zone to occupy a greater volume and space of soil for a higher nutrient uptake relative to roots; (2) under low P conditions, mycorrhizal fungi can potentially maximize carbon efficiency (defined as the amount of nutrient acquisition per unit carbon allocated belowground of plants) for nutrient acquisition as opposed to plants [123]. Therefore, employing mycorrhizal-mediated P acquisition makes economic sense for greater exploitation with the least cost. For instance, some earlier studies have indicated that approximately 4–20% of photosynthetically made C, transported to the root system, goes to fungi, potentially representing a massive loss to the plant [124,125,126].

### 5.3. Transport of Organic P Solubilizing Bacteria (PSB)

One crucial advantage of microbially mediated nutrient uptake is the ability of AM fungi to proliferate in organic P-rich patches. However, evidence suggests that the prospect of extensive proliferation in the organic P patches in response to spatial heterogeneity of soil nutrients strongly depends on the interaction between AM fungi and PSB [127]. The AM fungi and PSB are the key soil microbial functional groups whose activities directly influence P turnover and plant P uptake [127], particularly in heterogeneous microhabitats. A simple reason for such beneficial interaction is that, while AM fungi serve as a highway or a transporter of PSB [96], the PSB, in turn, increases the soil P via the release of phosphatase and organic acid, which facilitates organic P mineralization [96,128]. For example, Jiang et al. [96] recently reported that AM fungi transported phosphate-solubilizing bacteria to organic patches to facilitate the mineralization of soil organic P. Additionally, previous studies have shown that the PSB also stimulates hyphal growth [81], inhibits the growth of pathogenic diseases [129], and increases AM fungi fitness [130] and ecological functioning [104]. AM fungi can access discrete nutrient patches beyond the rhizosphere through this partnership by deploying its extensive hyphal networks.

### 5.4. Plant-Mycorrhizal Fungi Symbioses

Lastly, terrestrial ecosystem plants have evolved varying strategies to cope with soil-water deficits and mitigate its detrimental impacts [131], including plant-mycorrhizal fungi symbioses [96,115]. It is widely known that plant water retention is strongly impaired under drought conditions and can affect its metabolic function and fitness [132]. However, a prominent member of the soil community, such as AMF, has been shown to potentially modify the stomatal behavior and modulate root hydraulic conductivity [133], which ultimately plays a vital role in plant water retention, growth, and productivity. With the prospect of accessing soil water even below the permanent wilting point [134], AMF can improve the nutrition and osmoregulation of the host plant [131,133]. In a more recent study with ^18^O-labelled water, Kakouridis et al. [132] found that the amount transported via the AMF hyphal network accounted for 34.6% of the total water transpired by the host plant, suggesting that AMF can bridge air gaps in the soil matrix and access water beyond the reach of plant roots [132,135]. Therefore, among the multiple strategies employed by plants for efficient nutrient acquisition in heterogeneous and stressful environments, mycorrhizal symbioses could provide the highest benefit to plants.

## 6. Plants—Soil Microorganisms—Soil Nutrient Heterogeneity Loop

Plant ecologists have extensively considered the interaction between plants and their microbiomes in the last few decades. These studies span the individual species of plants [136,137] and soil microorganisms with a recent consideration of populations and community studies [60]. The soil microbial communities play critical roles in plant growth, which has a strong implication for ecosystem development and nutrient dynamics [58,59]. Similarly, plants exert some form of reciprocal effects on the function and biodiversity of the soil community. Most of the benefits derived from soil microorganisms and plants directly and indirectly affect growth and productivity. For example, accumulated evidence from previous studies has indicated that through the photosynthetic processes, plants supply the required C to the soil community in exchange for the most limiting nutrients [37]. Indeed, the exudates from plants and litter distribution are not uniformly arranged within the soil matrix, resulting in various scales of heterogeneity and a varying contrast of the soil nutrients.

Such spatial heterogeneity of soil nutrients has been extensively studied and has been found to promote plant growth, species composition, and coexistence with plant communities [28,29]. It is known that soil nutrient heterogeneity mediates intra-and inter-specific competitive interaction with neighboring plants [25]. However, the responses of plants to the effects of spatial heterogeneity of soil nutrients are most often species specific. A simple reason is that the scale and contrast of soil nutrients can influence soil heterogeneity [21,30], which determines the extent of sensitivity or the responses of plants to soil nutrient heterogeneity [48]. As a result, plants have developed various strategies and mechanisms to effectively uptake available soil nutrients.

However, the influence of soil microorganisms on plant responses to spatial heterogeneity of soil nutrients has not gained much attention in the scientific community. Therefore, more research studies are needed to bridge the gap and to increase our scope of understanding the potential roles of soil microorganisms on nutrient uptake, particularly under conditions of patchily distributed habitats. Soil nutrient availability is essential for plant growth and the belowground food web. The interactions of these three key factors (plant, soil microorganisms, and soil nutrient heterogeneity) converge to enhance ecosystem productivity (Figure 3).

So far, very few studies have attempted to highlight the impact of this three-way interaction on plant performance [13,65]. Therefore, more studies are needed to understand the response of plant individuals, populations, and plant communities in this interaction in both short- and long-term, as well as in field and controlled studies.

## 7. Conclusions and Prospects

We, therefore, conclude that soil microorganisms can potentially influence the spatial heterogeneity of soil nutrients on plant performance via the exploitation of patchy nutrients, organic matter mineralization, and the homogenization of patchy nutrients. By clarifying the role of soil microorganisms on the effect of soil nutrient heterogeneity on plant performance, the present review enhances the current understanding of ecosystem nutrient dynamics in the context of patchily distributed soil nutrients. More studies involving factors (e.g., climate change effects such as drought) that can alter the interactions of the plants, soil microorganisms, and soil nutrient heterogeneity are required to understand their responses to the ongoing global environmental change.

## Figures and Tables

**Figure 1 microorganisms-10-02399-f001:**
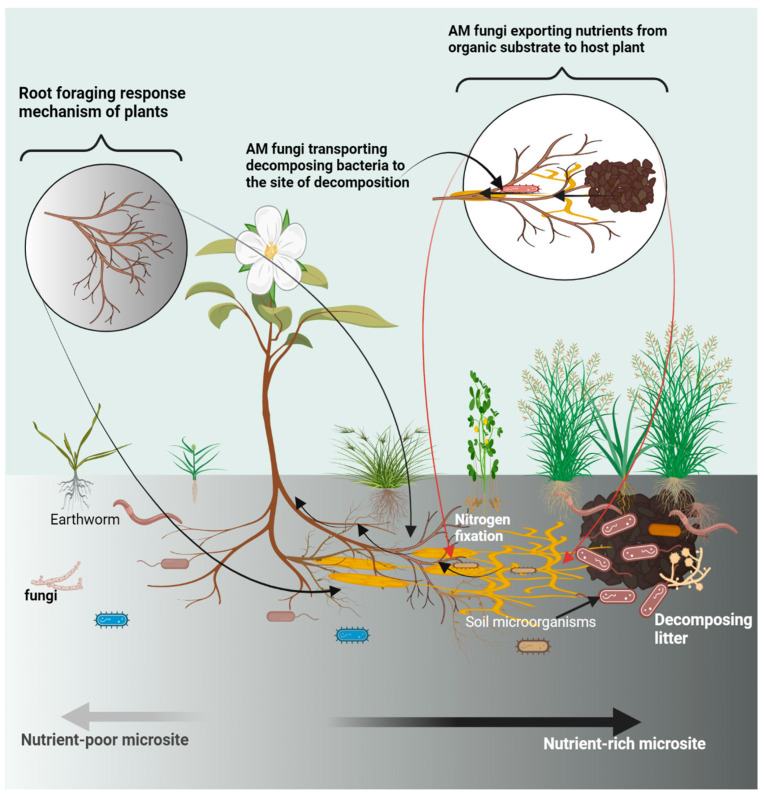
The role of soil microorganisms in regulating the effects of soil nutrient heterogeneity.

**Figure 2 microorganisms-10-02399-f002:**
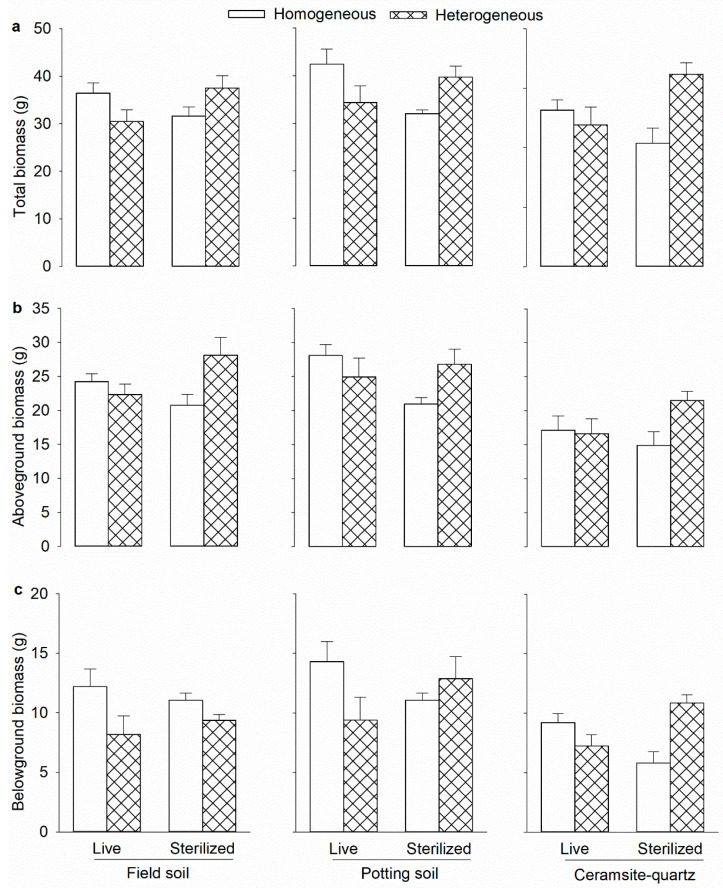
(**a**) Total, (**b**) aboveground, and (**c**) belowground mass of *Leymus chinensis* in the homogeneous or heterogeneous treatment in the presence of live or sterilized soil biota and each of the three soil substrates (field soil, potting soil, and ceramsite–quartz mixture substrates). Mean ± SE (*n* = 6) are given. See Table 2 for ANOVA results. Adapted with permission from Adomako et al. [13]. Copyright 2021, Springer Nature Switzerland AG 2021.

**Figure 3 microorganisms-10-02399-f003:**
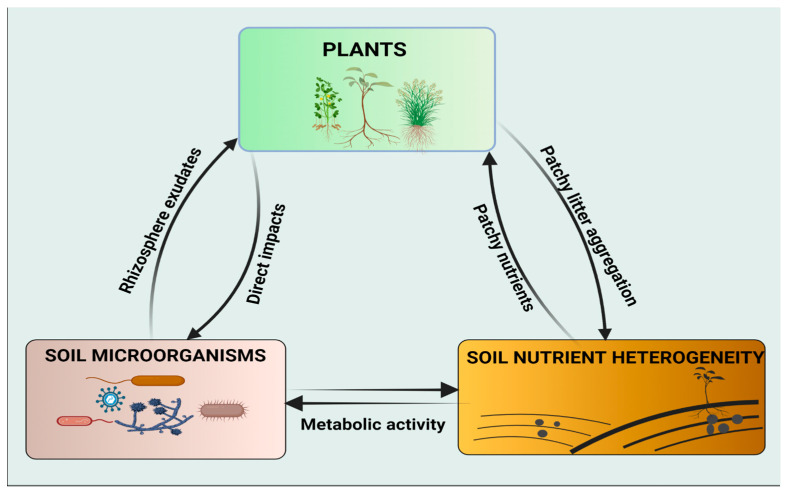
Schematic representation of the interactions between the plant, soil microorganisms, and soil nutrient heterogeneity. Both plants and soil microorganisms can acquire their nutrients from patchy microsites and alter soil properties by organic litter decomposition and metabolic activities, respectively. Soil microorganisms have diverse, direct effects on plants, e.g., mineralization of organic matter and homogenization of patchy nutrients. Plants interact with soil microorganisms through metabolites exuded by the roots, particularly in the rhizospheric zone.

**Table 1 microorganisms-10-02399-t001:** Summary of effects of soil microorganisms on plant performance/response in different heterogeneous substrates.

Identity of Soil Microorganism	Substrate Type	Host Plant	Effects of Soil Microorganism	References
*Glomus etunicatum*	Karst soil patches	*Biden pilosa*	Increased overall plant performance	[98]
Soil biota	Field, potting, and ceramsite-quartz.	*Leymus chinensis*	Decreased plant growth in heterogeneous but increased it in homogeneous soil.	[13]
Soil microorganisms	N-rich patches	*Lolium perenne*	Reduced N in microbial biomass.	[104]
*Gigaspora margarita*	Heterogeneous P.	*Trifolium subterraneum*	Increased total plant biomass and competitive intensity.	[105]
AM fungi species	Heterogeneous P.	*Linum usitatissimum*	Variable response among AM fungi species.	[106]
AM fungi	N-organic patches	*Lolium perenne*	No visible effects detected.	[107]
AM vs. EM fungi	Nutrient-rich patches	Multiple tree species	AM fungi increased root proliferation but EM decreased it.	[108]
AM vs. EM fungi	Nutrient-rich patches	Multiple tree species	High root and foraging precision and AM fungi proliferation.	[109]
*Hebeloma syrjense*	Nutrient-rich patches	Salix hybrid	Strong N-uptake from organic patches.	[110]

**Table 2 microorganisms-10-02399-t002:** Effects of soil substrates (field soil versus potting soil versus ceramsite–quartz mixture), soil heterogeneity (homogeneous versus heterogeneous), soil biota (live versus sterile), and their interaction effects on the total, aboveground, and belowground biomass of *Leymus chinensis* at the whole-pot level. See Figure 2 for data.

Effect	df	Total Mass		Aboveground Mass		Belowground Mass	
		*F*	*p*	*F*	*p*	*F*	*p*
Substrate (S)	2,60	**21.3**	**<0.001**	**16.5**	**<0.001**	**5.3**	**0.007**
Heterogeneity (H)	1,60	1.0	0.301	*3.5*	*0.065*	1.3	0.250
Soil biota (B)	1,60	<0.1	0.924	<0.1	0.948	0.2	0.614
S × H	2,60	1.4	0.239	0.5	0.608	3.2	0.046
S × B	2,60	0.3	0.730	0.8	0.444	0.2	0.768
H × B	1,60	**21.2**	**<0.001**	**11.0**	**0.001**	**13.6**	**<0.001**
S × H × B	2,60	0.5	0.557	<0.1	0.980	1.1	0.312

Degrees of freedom (df) and *F*- and *p*-values of three-way ANOVAs are presented. Values are in bold when *p* < 0.05 and in italics when 0.05 < *p* < 0.1.

## Data Availability

Not applicable.

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
