# Peer review of "Potential Roles of Soil Microorganisms in Regulating the Effect of Soil Nutrient Heterogeneity on Plant Performance"

_microorganisms, 2022, doi:10.3390/microorganisms10122399_

Round 1

Reviewer 1 Report

Very well-written paper. Although the literature was reviewed extensively I would suggest looking over the following:

Karst Soil Patch Heterogeneity with gravels promotes plant root development ....with arbuscular Mycorrhizal Fungi. Li et al 2022.

Fig. 3 is very well done but there is a typo in the figure. 

I missed more data, graphs, and tables showing more evidence. I also missed contrasting proof.

Author Response

Reviewer 1

Comments and Suggestions for Authors

Very well-written paper. Although the literature was reviewed extensively, I would suggest looking over the following:

Karst Soil Patch Heterogeneity with gravels promotes plant root development ...with arbuscular Mycorrhizal Fungi. Li et al. 2022.

Response: We read and cited the suggested article, i.e., Li et al. (2022), now ref. 98, (Lines 275 and 282 and in Table 1).

Fig. 3 is very well done, but there is a typo in the figure. 

Response: We corrected the typo in Figure 3.

I missed more data, graphs, and tables showing more evidence. I also missed contrasting proof.

Response: We added a new table, now table 1, summarizing previous studies that used various microorganisms to exploit nutrients from patchy soils or substrates (Line 288).

Reviewer 2 Report

Soil microorganisms is very important for plant nutrient uptake in the soil nutrient heterogeneity, thus further influence the plant performance in the changeable environment. The manuscript reviews the potential role of the soil microorganisms communities in regulating the effecs of soil nutrient heterogeneity on plant growth. They highlighted 1) the ecological importance of soil nutrient heterogeneity and soil microorganisms and discuss nutrient uptake mechanisms in heterogeneous soil, 2) the evolutionary advantages of nutrient uptake via the soil microbe in a heterogeneous soil, 3) a three-way interaction among the plant, soil nutrient heterogeneity, and soil microbes and propose areas for future research priorities. As we all know, the relationships among plant, soil, and microorganisms is complex and is difficult for understanding in diverse plant and soil types. Although, I suggest the authors should consider the functional plant species, and the global change (nitrogen deposition and precipitation change, et al.) in discussing the the role of soil microorganisms in plant nutrient uptake and transformation and the effects of soil microbes on plant performance. Lastly, I suggest the review can be accepted for publication after a minor revision. Thank you.

Author Response

As we all know, the relationships among plants, soil, and microorganisms are complex and difficult to understand in diverse plant and soil types.

Although I suggest, the authors consider the functional plant species and the global change (nitrogen deposition and precipitation change, et al.) in discussing the role of soil microorganisms in plant nutrient uptake and transformation and the effects of soil microbes on plant performance. Lastly, I suggest the review can be accepted for publication after a minor revision. Thank you.

Response: Thank you for prompting us. In fact, we discussed these global change effects in relation to soil nutrient heterogeneity and plant responses in the previous version of the manuscript (Lines 132 -145). We have also revised some portions in the text to consider how the soil microorganisms mediate the ongoing elevated nutrient depositions effects on plant performance (lines 421-428).

We also check the entire manuscript and corrected typos and other errors.